# EXPOSING TEXT-IMAGE INCONSISTENCY USING DIFFUSION MODELS

**Mingzhen Huang, Shan Jia, Zhou Zhou, Yan Ju, Jialing Cai, Siwei Lyu**
University at Buffalo, State University of New York

## ABSTRACT

In the battle against widespread online misinformation, a growing problem is text-image inconsistency, where images are misleadingly paired with texts with different intent or meaning. Existing classification-based methods for text-image inconsistency can identify contextual inconsistencies but fail to provide explainable justifications for their decisions that humans can understand. Although more nuanced, human evaluation is impractical at scale and susceptible to errors. To address these limitations, this study introduces D-TIIL (Diffusion-based Text-Image Inconsistency Localization), which employs text-to-image diffusion models to localize semantic inconsistencies in text and image pairs. These models, trained on large-scale datasets act as "omniscient" agents that filter out irrelevant information and incorporate background knowledge to identify inconsistencies. In addition, D-TIIL uses text embeddings and modified image regions to visualize these inconsistencies. To evaluate D-TIIL's efficacy, we introduce a new TIIL dataset containing 14K consistent and inconsistent text-image pairs. Unlike existing datasets, TIIL enables assessment at the level of individual words and image regions and is carefully designed to represent various inconsistencies. D-TIIL offers a scalable and evidence-based approach to identifying and localizing text-image inconsistency, providing a robust framework for future research combating misinformation. Please refer Project Page for source code and dataset.

## 1 INTRODUCTION

The widespread online misinformation (Ali, 2020) has become the bane of the Internet and social media. One simple means to create misinformation is to juxtapose images with texts that do not accurately reflect the image's original meaning or intention. In this work, we term this type of misinformation as text-image inconsistency (Lee & Choi, 2019; Tan et al., 2020; Zeng et al., 2023). Exposing text-image inconsistency has become an important task in combating misinformation. Text-image inconsistency can be solved with binary classification, as in the recent works of MAIM (Jaiswal et al., 2017), COSMOS (Aneja et al., 2021), NewsCLIPpings (Luo et al., 2021), and CCN (Abdelnabi et al., 2022), which classifies an input text-image pair as contextual consistent or inconsistent. Although showing good classification performance on benchmark datasets, the classification-based methods output only the predicted categories, with little or no evidence to support the decision. On the other hand, humans often spot text-image inconsistency by locating image regions corresponding to objects or scenes inconsistent with the textual description, using knowledge of the world. In addition, humans often prefer more visual evidence of semantic inconsistency, as when a mis-contextualized text-image pair is explained to another human. However, when we need to analyze many text-image pairs, relying on human inspection is costly, time-consuming, and prone to mistakes and errors (Molina et al., 2021). Our work aims to make this process automatic so it can scale up.

Specifically, we aim to address two challenges in localizing text-image inconsistency intrinsic to the complex nature of semantic contents across the two modalities. First, there is unrelated information in text and images irrelevant to their semantic consistency. This is usually information only represented in one modality but not in the other. Unrelated information in one modality will not have a counterpart in the other, but it is not the cause of semantic mismatch and cannot be accounted for inconsistency. Furthermore, many cases of inconsistency are hard to identify due to limited background knowledge of humans or algorithms. For instance, to someone who is unaware that dolphins are mammals, a text stating "a school of fish swimming in the ocean" might seem consistent with an image showing

dolphins swimming. Such missing information can be overcome by using a more knowledgeable human or incorporating background knowledge into the algorithm.

Both challenges are addressed in our work by leveraging the text-to-image diffusion models. Text-to-image diffusion models trained on large-scale datasets, such as DALL-E2 (Ramesh et al., 2022), Stable Diffusion (Rombach et al., 2022), Glide (Nichol et al., 2021), and GLIGEN (Li et al., 2023), can generate realistic images with consistent semantic content in the text prompts. We can regard these large-scale text-to-image diffusion models as an "omniscient" agent with extensive background knowledge about any subject matter. Taking advantage of this knowledge

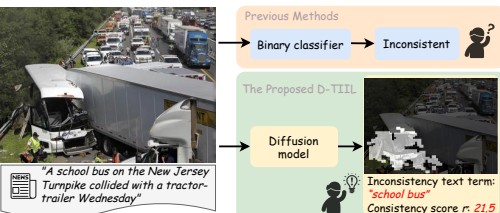

Figure 1: *Exposing text-image inconsistency based on previous methods and our method. Instead of employing a binary classification model, D-TIIL offers interpretable evidence by localizing word- and pixel-level inconsistencies and quantifying them through a consistency score.*

representation, we describe a new method that *locates* the semantically inconsistent image regions and words, which is termed as Diffusion-based Text-Image Inconsistency Localization (D-TIIL). D-TIIL proposes two different alignment steps that iteratively align the image and update text (in the form of vectorized embeddings) with diffusion models to (i) filter out the irrelevant semantic information in the text-image pairs and (ii) incorporate background knowledge that is not obvious in their shared semantic scope. The first alignment step employs diffusion models to generate aligned text embeddings from the input image. This is to filter out implicit semantics and establish textual consistency. The second alignment operation focuses on denoising the input text to be more relevant to the input image. To be more specific, we modify the input image based on the original input text to achieve semantic consistency, and then produce aligned text embeddings from this edited image. These two alignment steps yield modified yet knowledge-shared text embeddings from both the input image and text, making it easier to identify semantic inconsistencies. This approach sets our method apart from previous ones, as depicted in Fig. 1.

Existing datasets (Luo et al., 2021; Aneja et al., 2021) do not provide evidence of inconsistency at the level of image regions and words that can be used to evaluate D-TIIL. To this end, we create a new Text-Image Inconsistency Localization (TIIL) dataset, which contains 14K text-image pairs. Existing datasets construct inconsistent text-image pairs by randomly swapping texts (Luo et al., 2021) or external search for the similar text of the swapped texts (Aneja et al., 2021) to match with the original image. These methods can create inconsistent text-image pairs that either conflict with human intuitions or are totally irrelevant (see more analysis in Sec.4). Differently, TIIL constructs inconsistent pairs by changing words in the text, and/or editing regions in the image (*e.g.*, changing objects, attributes, or scene-texts)*. The edited words and regions are manually selected. The image editing is made with the text-to-image diffusion models. Furthermore, all inconsistent text-image pairs undergo meticulous manual curating to reduce ambiguities in interpretation.

The main contributions of our work can be summarized as follows:

- We develop a new method, D-TIIL, that leverages text-to-image diffusion models to expose text-image inconsistency with the location of inconsistent image regions and words;
- Text-to-image diffusion models are used as a latent and joint representation of the semantic contents of the text and image, where we can align text and image to discount irrelevant information, and we use the broad coverage of knowledge in the diffusion models to incorporate more extensive background;
- We introduce a new dataset, TIIL, built on real-world image-text pairs from the Visual News dataset, for evaluating text-image inconsistency localization with pixel-level and word-level inconsistency annotations.

## 2 BACKGROUNDS

### 2.1 RELATED WORKS

Text-image inconsistency detection has been the focus of several recent works (Abdelnabi et al., 2022; Luo et al., 2021; Qi et al., 2021; Aneja et al., 2021; Abdelnabi et al., 2022). There are many methods (Zlatkova et al., 2019; Abdelnabi et al., 2022; Popat et al., 2018) using the reverse

---

*TIIL has mixed both original and edited images; one cannot simply rely on forensic methods that identifying image editing to expose inconsistent pairs.

image search function provided by the Search Engine (*e.g.*, Google Image Search) to gather textual evidence (articles or captions) from the Internet or external fact-checking sources on the Internet (e.g., Politifact[†] and Factcheck[‡]). The resulting text is then compared with the original text in an embedding space such as BERT (Kenton & Toutanova, 2019) to determine their consistency. Although straightforward and intuitive, these methods rely solely on the results from the reverse image search and are limited by irrelevant or contradicting texts found online. Other inconsistency detection methods explore joint semantic representations of texts and images. For example, Khattar et al. (2019) designs a multimodal variational autoencoder for learning the relationship between textual and visual information for fake news detection. McCrae et al. (2021) detect semantic inconsistencies in video-caption posts by comparing visual features obtained from multiple video-understanding networks and textual features derived from the BERT (Kenton & Toutanova, 2019) language model. Aneja et al. (2021) employs a self-supervised training strategy to learn correlation from an image and two captions from different sources. More recently, neural vision-language models originally designed for other vision-language tasks (e.g., VQA, image-text retrieval) have also been applied to text-image inconsistency detection. For instance, CLIP (Radford et al., 2021b) is used in Luo et al. (2021) and the VinVL (Zhang et al., 2021) model introduced in Huang et al. (2022).

Several multi-modal datasets exist for image inconsistency detection. MAIM (Jaiswal et al., 2017), MEIR (Sabir et al., 2018), FacebookPost (McCrae et al., 2022) and COSMOS (Aneja et al., 2021) are formed by swapping the original caption of an image with randomly selected ones to create inconsistent image-text pairs. The NewsCLIPpings dataset (Luo et al., 2021) utilizes CLIP (Radford et al., 2021a) as a retrieve model to swap similar captions in the Visual News dataset (Liu et al., 2020). The main problem with these datasets is that the semantic relations among the labeled consistent or inconsistent pairs are not precise (Huang et al., 2022), making them less reliable to be used as training data for text-image inconsistency detection methods. These methods and datasets do not report words or image regions that cause the inconsistency.

## 2.2 TEXT-TO-IMAGE DIFFUSION MODELS

The diffusion model (Ho et al., 2020) has recently attained state-of-the-art performance in the field of text-to-image generation (Ramesh et al., 2022; Saharia et al., 2022; Rombach et al., 2022). As a category of likelihood-based models (Nichol & Dhariwal, 2021), diffusion models perturb the data by progressively introducing Gaussian noise to the input data and train to restore the original data by reversing this noise application process. The key idea involves initialization with $\mathbf{x}_T \sim \mathcal{N}(0, \mathbf{I})$, which represents an iteratively noised image derived from the input image $\mathbf{x}_0$. At each timestep $t \in [0, T]$, the sample $\mathbf{x}_t$ is computed as $\mathbf{x}_t = \sqrt{\alpha_t}\mathbf{x}_0 + \sqrt{1 - \alpha_t}\boldsymbol{\epsilon}_t$, where $\alpha_t \in (0, 1]$ defines the level of noise, and $\boldsymbol{\epsilon}_t \sim \mathcal{N}(0, \mathbf{I})$ represents the sampled noise. Ultimately, the distribution of $\mathbf{x}_T$ approaches a Gaussian distribution. Diffusion models then iteratively reverse this process and denoise $\mathbf{x}_T$ to generate images given a text conditioning $c$ by minimizing a simple denoising objective:

$$\mathcal{L} = \mathbb{E}_{t, \mathbf{x}_0, \boldsymbol{\epsilon}} \left\| \boldsymbol{\epsilon} - \epsilon_\theta \left( \mathbf{x}_t, t, c \right) \right\|_2^2 \tag{1}$$

where $\epsilon_\theta$ is an UNet (Ronneberger et al., 2015) noise estimator that predicts $\epsilon_t$ from $\mathbf{x}_t$. Diffusion models have been widely used in various downstream applications, including image editing (Couairon et al., 2023; Kawar et al., 2023) where a text-conditional diffusion model can be generalized for learning conditional distributions. When provided with different text conditionings, the model generates different noise estimates. Notably, the variation in noise across spatial locations reflects the semantic distinctions between the corresponding text conditions in the image space. This inspiration motivates us to utilize diffusion models for representing and exposing semantic inconsistencies in image inconsistency.

## 3 METHOD

This section describes the proposed D-TIIL model in detail. The input to D-TIIL is a pair of image $I$ and text $T$, from which D-TIIL outputs the image region (as a binary mask $\mathcal{M}$) and words in the text that exhibit semantic inconsistency. In addition, a consistency score $r \in [0, 100]$, with 0 being maximum inconsistent and 100 being completely consistent, is also obtained based on the localization results. The overall process of D-TIIL is illustrated in Fig. 2 with four distinct steps. In those four steps, we iteratively align image-text semantic to produce final output as shown in Fig. 3

---

[†]Politifact: https://www.factcheck.org
[‡]Factcheck: https://www.politifact.com

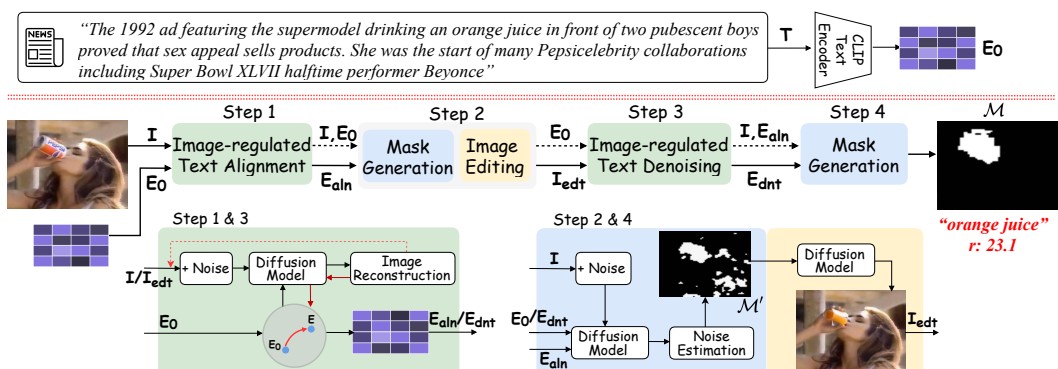

Figure 2: *The overall pipeline of D-TIIL. See texts for details.*

**Step 1: Align Text Embedding to Input Image.** We first obtain the CLIP (Radford et al., 2021a) text embedding $E_0$ of the input text $T$. Using a pre-trained Stable Diffusion model $\mathcal{G}$ (Rombach et al., 2022), we find another text embedding $E_{aln}$ that better aligns with the semantic content of the input image $I$ as shown in Fig. 3 Step 1. Model $\mathcal{G}$ takes as input the text embedding $E_0$ and noised image $\mathbf{x}_T$ to generate an image, $\mathcal{G}(\mathbf{x}_T; E_0)$. It simulates a diffusion process that begins with the input noise and generates an image that exhibits similar semantic content to the text embedding $E_0$. The image-regulated text alignment in D-TIIL is to solve the following optimization problem:

$$E_{aln} = \arg\min_E \|I - \mathcal{G}(\mathbf{x}_T; E)\|_2 \quad s.t., \|E - E_0\|_F \leq \gamma \tag{2}$$

where the learnable $E$ is initialized from $E_0$, and $\gamma > 0$ is a small constant determined as a hyper-parameter. $\|\cdot\|_2$ and $\|\cdot\|_F$ are the vector $\ell_2$ norm and matrix Frobinuous norm, respectively. The constraint is used to control the deviation from the original embedding. This optimization problem can be solved using the gradient computation of $\mathcal{G}$ iteratively. The obtained $E_{aln}$ is semantically closer to the image when the original text $T$ has inconsistencies and distracting semantic information.

**Step 2: Text-guided Image Editing.** Next, we generate an edited image, $I_{edt}$, which is in alignment with the original text embedding, $E_0$. This is intended to materialize the original text, $T$, within a visual context, thereby minimizing the presence of extraneous and implicit data. It subsequently acts as the benchmark for estimating inconsistency. D-TIIL transposes the semantics of $E_0$ into the image space, with both $E_0$ and the aligned text embedding $E_{aln}$ guiding the editing process. Specifically, we introduce a noised version of image $I$ as the input and leverage the UNet architecture from the Stable Diffusion to derive two noise estimations, each corresponding to $E_0$ as target and $E_{aln}$ as reference. By examining the difference between these two noise estimates in the spatial domain, we can identify regions in image $I$ that are most prone to modifications due to the shift in conditioning text from $E_{aln}$ to $E_0$. This disparity is then transformed into a binary mask, denoted as $\mathcal{M}'$, by normalizing values within the [0, 1] range and subsequently employing a thresholding operation. After obtaining $\mathcal{M}'$, we use the Diffusion Inpainting model (Lugmayr et al., 2022) to yield the edited image $I_{edt}$, guided by the target text embedding $E_0$ and mask $\mathcal{M}'$. The process effectively transmutes the textual embedding $E_0$ into a visual equivalent, purging any distracting or implicit details, as shown in Fig. 3 Step 2.

**Step 3: Align Text Embedding to Edited Image.** While the binary mask $\mathcal{M}'$ captures the regions of inconsistency between the input image $I$ and text embedding $E_0$, it may still include regions that are not directly related to semantic consistency, such as the objects or scenes in $I$ and $I_{edt}$ that does not correspond to a verbal description in $T$. This is a form of unrelated information that we use another round of operation involving the diffusion model $\mathcal{G}$ to reduce. Specifically, we formulate another optimization problem $E_{dnt} = \arg\min_E \|I_{edt} - \mathcal{G}(\mathbf{x}_T^{edt}; E)\|_2$, s.t., $\|E - E_0\|_F \leq \gamma$ where $\mathbf{x}_T^{edt}$ is a noised image of $I_{edt}$. Compared with input text embedding $E_0$, the aligned text embedding $E_{dnt}$ includes extra implicit information from the images and excludes additional implicit information that only appears in the text as shown in Fig. 3 Step 3 where we refer this operation as image-regulated text denoising.

**Step 4: Inconsistency Localization and Detection.** From the two text embeddings that are more closely aligned, $E_{aln}$ and $E_{dnt}$, we generate the difference in visual domain denoted as $\mathcal{M}$. This is done by repeating the mask generation process outlined in Step 2. The $\mathcal{M}$ represents the pixel-level inconsistent region within the image. To detect the corresponding inconsistent words, we compare the

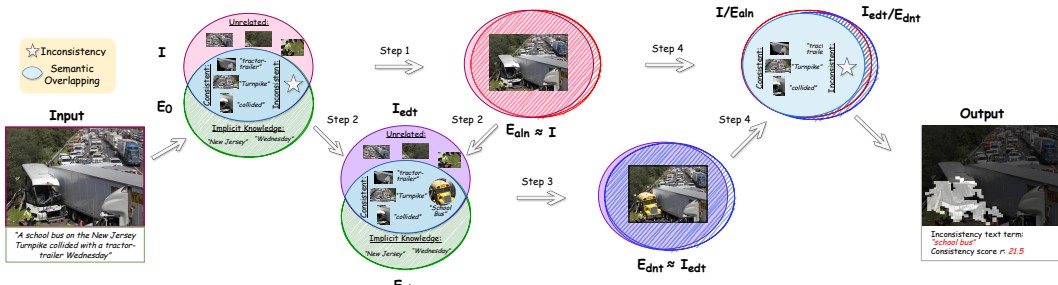

Figure 3: *The main process of D-TIIL is illustrated conceptually with Venn diagrams, where the semantic contents of text and image are represented as two circles. The four steps gradually align the semantic contents to facilitate exposure of inconsistency: given an initial image-text pair $(I, E_0)$, the proposed method first produces a text embedding $\mathbf{E}_{aln}$ that is aligned with $I$, and then an edited image $I_{edt}$ to filter the inconsistency. In Step 3, the model optimizes $E_0$ from the $I_{edt}$ to obtain a $E_{dnt}$ which is aligned with $I_{edt}$. Finally, in Step 4, the model produces the inconsistency mask from well-aligned pair $(I, E_{aln}, E_{dnt})$.*

edited image $I_{edt}$ with the inconsistency mask $\mathcal{M}$. Specifically, we leverage the CLIP image encoder to obtain the image embedding from the image $I_{edt}$. Next, we derive tokenized words from the rows in $E_0$ that exhibit the greatest cosine similarity with the image embedding, such as the example of "an orange juice" depicted in Fig. 2. To further generate a regression score that quantifies the degree of image-text inconsistency, we extract the CLIP image embedding from the masked input image $I$ using $\mathcal{M}$. We then compute the cosine similarity score between this CLIP image embedding of the masked image and the input text embedding $E_0$ as the consistency score. The resulting score is rescaled to the range of [0, 100], serving as the final consistent score $r$ of our D-TIIL model.

## 4 TIIL DATASET

We also construct TIIL as a more carefully curated dataset for text-image inconsistency analysis. Our approach to dataset creation is different from those of existing datasets that use randomly or algorithmically identified pairs as mismatched image-text pairs (Jaiswal et al., 2017; Sabir et al., 2018; Aneja et al., 2021). We leverage state-of-the-art text-to-image diffusion models to design text-guided inconsistencies within images and human annotation to improve the relevance of the inconsistent pairs.

**Data Generation.** Our methodology starts with a real-world image-text pair, $\{I, T\}$, obtained from the Visual News dataset (Liu et al., 2020), which offers a rich variety of news topics and sources, providing us with diverse real-world news data. The first step in our process involves creating an edited image, $I_e$. This is achieved by modifying a specific region in $I$ using an altered text $T_m$ by human annotators, where the text prompt corresponding to the object is replaced with a different generation term. The region to be manipulated is manually selected. Through this procedure, we can generate two consistent image-text pairs, namely $\{I, T\}$ and $\{I_e, T_m\}$, as well as two inconsistency pairs formed by $\{I, T_m\}$ and $\{I_e, T\}$. The DALL-E2 model (Ramesh et al., 2022), known for its capacity to generate images within specific regions based on text prompts, is leveraged to create this dataset. Table 1 demonstrates that the DALL-E2 model outperforms real-world image-text pairs in terms of CLIP similarity scores, indicating its superior ability to capture multi-modal connections. Fig. 4 illustrates the entire generation pipeline of our TIIL dataset. The process starts with a real image-text pair. Human annotators then identify the corresponding visual region and textual term. Subsequently, these annotators provide a different text prompt to replace the chosen term, thereby creating an inconsistency with the selected object region. The mask of the selected object region and the swapped text with the new prompt are then fed into the DALL-E2 model for image generation. In the final step, human annotators carefully assess the quality of the generated images, evaluate the image-text inconsistency, and refine the region mask to provide the ground truth for the pixel-level inconsistency mask. The TIIL dataset consists of approximately 14K image-text pairs, encompassing a total of 7,138 inconsistencies and 7,101 consistent pairs. All inconsistent instances in the dataset have been manually annotated.

**Manual Annotations.** We also go through a manual meticulous data annotation process by a team of six professional annotators. The annotation process is carried out following a defined procedure. First, the annotators select object-term pairs that align with each other, and then, they input the target text prompt that corresponds to the selected object-term pairs. The final step of the process involves

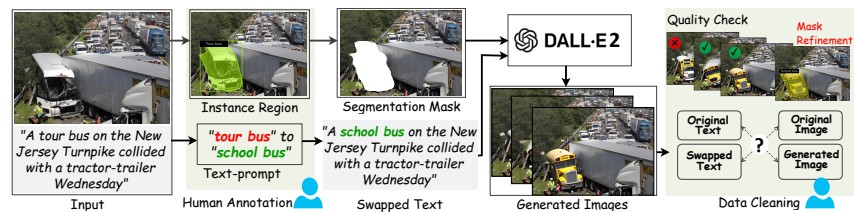

Figure 4: *Pipeline depicting the generation and annotation process of the proposed TIIL dataset.*

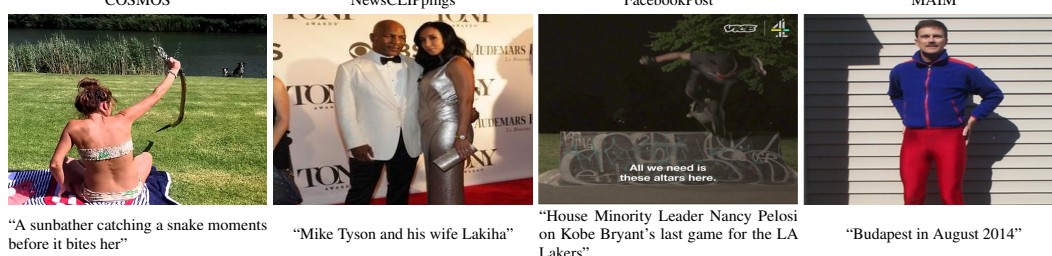

Figure 5: *Inconsistent examples from other datasets. Due to the constraints of using random swapping or auto-retrieval methods to produce inconsistent pairs, the resulting pairs could either be semantically consistent (as seen in the left two examples) or entirely unrelated (as illustrated by the right two examples).*

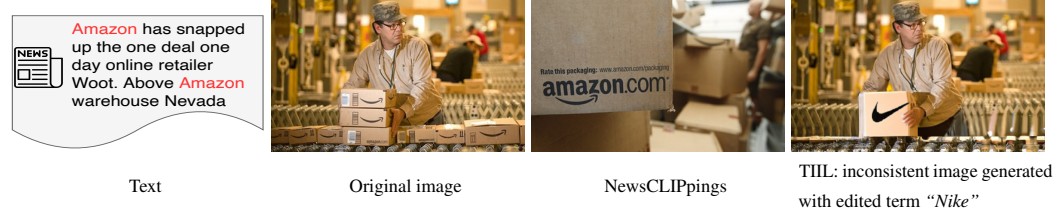

Figure 6: *An inconsistent example in our TIIL and NewsCLIPpings. Note that the TIIL example agrees with a common viewer to be* inconsistent *while the one from NewsCLIPpings is consistent rather than inconsistent.*

|  | Avg. CLIP Score |
|---|---|
| Real-world | 30.71 |
| DALL-E2 | 31.03 |
| Random Swap | 12.39 |
| Ours-Inconsistent | 22.68 |

Table 1: *Comparison of the CLIP scores for different pairs.*

| Dataset | Total #Pairs | Inconsistent #Pairs | Anno #Pairs | Generation Method | Pixel&word-level Annotation |
|---|---|---|---|---|---|
| MAIM | 240K | 120K | 0 | Random swap | × |
| MEIR | 58K | 29K | 0 | Random swap | × |
| NewsCLIPpings | 988K | 494K | 0 | Auto retrieval | × |
| FacebookPost | 4K | 2K | 0 | Random swap | × |
| COSMOS | 200K | 850 | 1.7K | Manual | × |
| TIIL (Ours) | 14K | 7K | 14K | Manual+Diffusion | ✓ |

Table 2: *Comparison with existing related datasets.*

data cleaning to ensure the accuracy and coherence of the dataset. To maintain the highest quality, the annotations are cross-validated among the team members. This step allows for the detection and rectification of any potential errors or inconsistencies. Further details about our data annotation process are in the Supplementary Material.

**Comparison with Existing Datasets.** As shown in Table 2, our TIIL dataset boasts several unique characteristics that set it apart from existing datasets. To the best of our knowledge, it is the first of its kind to feature both pixel-level and word-level inconsistencies, offering fine-grained and reliable inconsistency. We provide a comparison of the CLIP scores between the traditional random swap-based creation method and our diffusion-based approach in Table 1. The results demonstrate the superiority of our method in achieving higher levels of semantic similarity compared to randomly swapped image-text inconsistency pairs. This higher semantic similarity makes the inconsistencies more subtle and harder to detect, thereby enhancing the dataset's complexity and realism. In comparison to datasets that build inconsistency pairs through random swap (e.g., MAIM (Jaiswal et al., 2017), MEIR (Sabir et al., 2018), FacebookPost (McCrae et al., 2022) and COSMOS (Aneja et al., 2021)) or automatic retrieval (such as NewsCLIPping (Luo et al., 2021)), TIIL offers more reliable consistent and inconsistent pairs, as demonstrated in Fig. 5 and Fig. 6. Moreover, our dataset is sourced from a wide array of news topics and sources, ensuring a diverse and rich collection of examples. For a more comprehensive understanding, we present examples of image-text pairs and annotated labels from the TIIL dataset in Fig. 7. These examples underline the range and complexity of the data within our novel dataset.

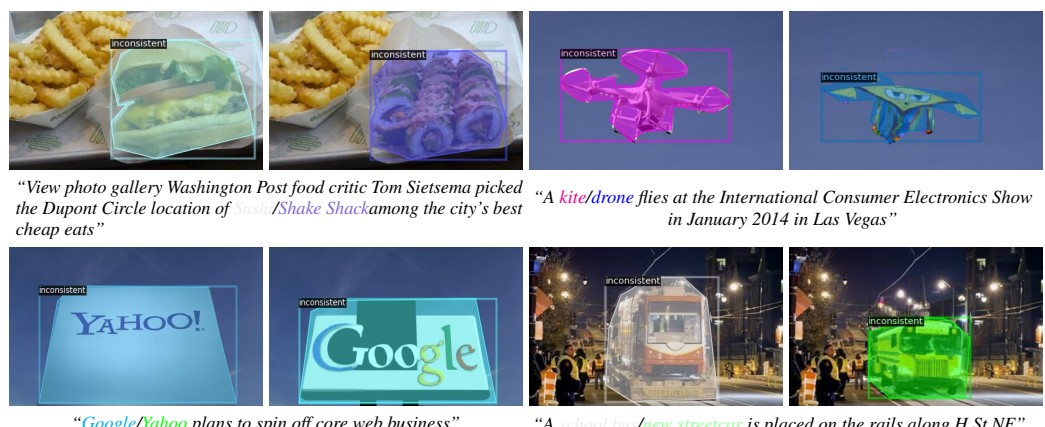

*"View photo gallery Washington Post food critic Tom Sietsema picked the Dupont Circle location of Sushi/Shake Shack among the city's best cheap eats"*

*"A kite/drone flies at the International Consumer Electronics Show in January 2014 in Las Vegas"*

*"Google/Yahoo plans to spin off core web business"*

*"A school bus/new streetcar is placed on the rails along H St NE"*

Figure 7: *Examples in TIIL dataset. Colored texts correspond to inconsistent regions of the same color in the image. The figure is best viewed in color.*

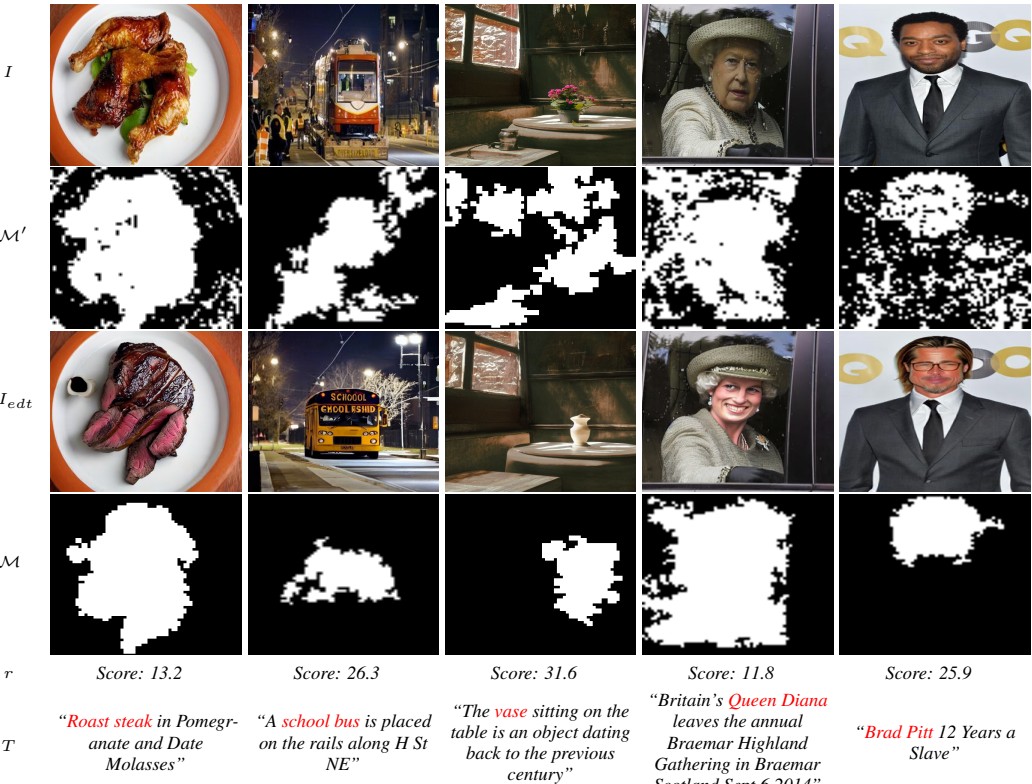

| Score: 13.2 | Score: 26.3 | Score: 31.6 | Score: 11.8 | Score: 25.9 |

*"Roast steak in Pomegranate and Date Molasses"*

*"A school bus is placed on the rails along H St NE"*

*"The vase sitting on the table is an object dating back to the previous century"*

*"Britain's Queen Diana leaves the annual Braemar Highland Gathering in Braemar Scotland Sept 6 2014"*

*"Brad Pitt 12 Years a Slave"*

Figure 8: *D-TIIL on TIIL examples. The detected inconsistent words are highlighted in red.*

## 5 EXPERIMENTS

This section presents a comprehensive analysis of our approach, including qualitative and quantitative results, comparisons with other methods, and ablation studies to evaluate different variations.

### 5.1 SETTINGS

**Implementation Details**. We use the implementations of Stable Diffusion (Rombach et al., 2022) and CLIP (Radford et al., 2021a) ViT-B/32 model available on https://huggingface.co. In the diffusion model, we use the denoising diffusion implicit model (DDIM) (Song et al., 2020) to sample noises, and classifier-free guidance (Ho & Salimans, 2022) is set to the recommended value of 7.5. We train both text embedding $E_{aln}$ and $E_{dnt}$ for 500 iterations with a learning rate of $4e^{-6}$. The hyperparameter $\gamma$ is set to 8 in our experiment. For noise estimation, we use the same random seed for two conditioned text embeddings, remove outlier values in noise predictions, and average spatial

|  | DetCLIP | GAE | Ours |
|---|---|---|---|
| mIoU (%) | 14.24 | 27.35 | 47.12 |

Table 3: *Comparison of text-image inconsistency localization.*

|  | CLIP | CLIP* | CCN | Ours |
|---|---|---|---|---|
| AUC (%) | 84.18 | 78.58 | 82.80 | **87.00** |
| Accuracy (%) | 76.77 | 71.87 | 74.10 | **79.46** |

Table 4: *Comparison of detection.*

differences over a set of 10 input noises. After obtaining the predicted inconsistent masks, we use a threshold to binarize them where the threshold is the average values among the mask. We only retain the top 3 mask regions with the largest areas. To localize inconsistent words, we follow the previous work (Radford et al., 2021a) to use a prompt template "A photo of {words}" for CLIP (Radford et al., 2021a) text embedding generation.

**Evaluation Metrics.** We report the mean of class-wise intersection over union (mIoU) (Everingham et al., 2015) to evaluate the quality of the predicted inconsistency mask. mIoU is a metric that aligns with the per-pixel classification formulation, making it a commonly used standard metric in semantic segmentation tasks (Fan et al., 2021; Klingner et al., 2021; Gao et al., 2022; Xu et al., 2022).

## 5.2 COMPARISON WITH EXISTING METHODS

We first present qualitative results demonstrating the pixel-level and word-level detection of inconsistency achieved by the proposed D-TIIL model in Fig. 8. It can be observed that our D-TIIL achieves accurate results benefiting from the multi-step semantic alignment. Given the absence of prior work especially in addressing image inconsistency localization, we consider the following two relevant baseline approaches for comparison: (1) a straightforward solution that uses an object detector (Zhou et al., 2022) to detect all objects as segmentation mask in the image and then compares the CLIP (Radford et al., 2021a) embedding similarity between the text and each object region. The object region with the highest dissimilarity is identified, and its corresponding mask is considered as the inconsistent mask. We denote this method as DetCLIP; (2) an off-the-shelf method, GAE (Chefer et al., 2021), which provides explainability for bi-modal and encoder-decoder transformers by presenting co-attention maps. Specifically, it analyzes the classification relevancy of specific layers for CLIP (Radford et al., 2021a) to provide a pixel-level attention heatmap. We further generate the inconsistent mask with the attention heatmap by applying a threshold. The comparison results on the TIIL dataset are shown in Table 3 and Fig. 9. The D-TIIL method demonstrates significant superiority over the baseline methods in both mIoU scores and qualitative evaluation.

## 5.3 ABLATION STUDIES

**Text-image Inconsistency Detection.** While we have emphasized that binary classification may not be the best method for revealing inconsistencies in text image pairs, we have adapted the D-TIIL model into a binary framework. This allows us to compare D-TIIL with current text-image inconsistency detectors, such as the CLIP model (Radford et al., 2021a), its fine-tuned version on NewsClippings (referred to as CLIP*), and a recent detector that shows the best performance CCN (Abdelnabi et al., 2022). We report the Area Under ROC (AUC) and Accuracy. As shown in Table 4, the D-TIIL model outperforms other models in terms of both AUC and Accuracy scores. This improvement can be attributed to the use of the detected inconsistency mask for image embedding extraction, which enables the exclusion of distracting and implicit regions from the image, thereby enhancing the classification performance. Moreover, compared with CCN, which requires inverse searches for inconsistency detection, our D-TIIL does not require information from external sources, especially showing superior performance on pairs with manipulated images that cannot be found on the Internet. Specifically, the experimental results demonstrate that CCN achieves an accuracy of 80.12% on the subset composed entirely of original images/text sourced from the Internet, However, its accuracy drops to 68.0% for subsets containing manipulated image/text. In comparison, our method obtains an accuracy of 82.79% and 79.15% on two subsets, respectively, indicating minimal impact on the online accessibility of data.

**Image-regulated Text Denoising.** We first highlight the benefits of performing additional image-regulated text denoising in Step 3 instead of using the $\mathcal{M}'$ in Step 2 as the inconsistency map. Table 5 and Fig. 8 reveal that $\mathcal{M}'$ exhibits a coarse pixel-level inconsistency map, whereas the detected inconsistency mask includes additional background areas due to the absence of denoising in the input text embeddings.

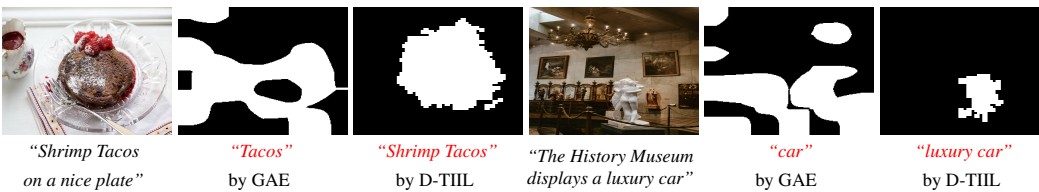

*"Shrimp Tacos*     *"Tacos"*     *"Shrimp Tacos"*     *"The History Museum*     *"car"*     *"luxury car"*
*on a nice plate"*   by GAE     by D-TIIL     *displays a luxury car"*   by GAE     by D-TIIL

Figure 9: *Comparison with GAE for detecting inconsistent mask.*

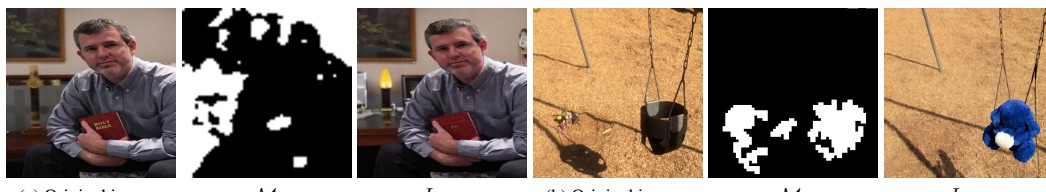

(a) Original image     $\mathcal{M}$     $I_{edt}$     (b) Original image     $\mathcal{M}$     $I_{edt}$

*Text (a): "Rev Nace Lanier poses for a portrait in his office at*     *(b): "A stuffed animal and flowers rest near a white swing in*
*Ronald Reagan National Airport"*     *Wills Memorial Park in La Plata MD where JiAire died last May"*

Figure 10: *Failure cases of D-TIIL on TIIL dataset. (a) is a consistent image-text pair and (b) is an inconsistent image-text pair. Detected inconsistent words are highlighted in* red.

**Text Embedding Alignments.** We further compare our method with two variations to show the influence of the text embedding alignment process on inconsistency localization. This experiment is conducted in a randomly sampled subset of TIIL with 1000 image-text pairs. We consider two initialization variations, (1) random initialization of $E_0$ from random noise; and (2) initialize $E$ with $E_0$ and is only supervised by the reconstruction $\mathcal{L}_2$ loss without the text embedding constrain loss in Eq. (2). Table 6 demonstrates that the performance decreases when the aligned text differs from the initial text embedding $E_0$ too much.

## 5.4 FAILURE CASES

Fig. 10 shows two examples that D-TIIL does not generate results consistent with human viewers. Such cases are attributed to the limitations of the underlying text-to-image diffusion models in understanding the entailment of semantic meanings of words and creating precise local edits of images. In Fig. 10 (a), the word "office" is likely to be taken too literally by the model, even though a human viewer may extrapolate to this unusual setting. The case of Fig. 10 (b) is a clear inconsistent pair, as the word "stuffed animal" is not reflected in the original image, but D-TIIL finds misaligned inconsistent regions.

## 6 CONCLUSION

In this work, we describe D-TIIL to expose text-image inconsistency by employing diffusion models as an omniscient, impartial evaluator to learn the semantic connections between textual and visual information. Instead of using a binary classification model, D-TIIL offers interpretable evidence by determining the inconsistency score of an image-text pair and pinpointing potential areas where the text and image semantics disagree. We also provide a new dataset, TIIL, built on real-world image-text pairs, for evaluating our D-TIIL method. Experimental evaluations of D-TIIL on this dataset demonstrate improved and more explainable results than the previous methods. There are a few directions we would like to enhance the current work in the future. Given the limited prior knowledge of the diffusion model we used, our model may not effectively handle the inconsistencies with respect to specific external knowledge. One potential solution is to replace our general foundation diffusion model with domain-specialized diffusion models to learn semantic connections specific to those domains (e.g., a text-to-image diffusion model trained on a Fashion dataset (Sun et al., 2023; Karras et al., 2023) that generates fashion images would identify inconsistencies in fashion-related text-image pairs such as mismatched brands or styles). Furthermore, it is important to continue to enlarge our dataset with more recent text-prompt image generation models.

|  | $\mathcal{M}'$ | $\mathcal{M}$ |
|---|---|---|
| mIoU (%) | 38.43 | 47.12 |

Table 5: *Benefits of text denoising.*

|  | Random Init | w/o Emb Constrain | Ours |
|---|---|---|---|
| mIoU (%) | 41.27 | 43.47 | 47.50 |

Table 6: *Comparison of text embedding alignments.*

**Ethics Statement**. This work is relevant to the fight against misinformation, which is a vexing problem that reduces the integrity of online information. While our method can effectively expose misinformation created with text-image inconsistency, there is a risk that the misinformation creator may use our method to select more deceptive text and image pairs, for instance, only use those that pass our algorithm. The mitigation to such abuse is to continue improving the algorithm and only provide its access to trustworthy parties. We will only release our code as open-source with the condition that it must not distribute harmful, offensive, dehumanizing content or otherwise harmful representations of people or their environments, cultures, religions, etc. produced with the model weights.

**Acknowledgement**. This work was supported in part by the US Defense Advanced Research Projects Agency (DARPA) Semantic Forensic (SemaFor) program, under Contract No. HR001120C0123, and National Science Foundation (NSF) Project SaTC-2153112. The views and conclusions contained herein are those of the authors and should not be interpreted as necessarily representing the official policies, either expressed or implied, of DARPA, NSF, or the U.S. Government.

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

APPENDIX

In the Appendix, we provide more details about TIIL dataset and conduct additional ablation experiments.

## A  TIIL DATASET

We provide more annotation details and statistics about the TIIL dataset in this section. More examples in the dataset are shown in Fig. 11.

**Annotation Details.** The annotation process was carried out by a team of six annotators with professional background, including 1 postdoc from the research team, 3 graduate volunteers, and 2 undergraduate volunteers. These annotators possess a comprehensive understanding of the data annotation task, which comprised three steps: (1) selecting matched object-term pairs and inputting the target text prompt, (2) manipulating the image or text with exact instruction and (3) conducting data cleaning as the final step. The generation of the TIIL dataset starts with a real image-text pair. The first annotation process involves identifying the corresponding visual regions and textual terms through human involvement. Initially, we automatically extract separate text terms with spaCy [§], human annotators then select and annotate the visual region that corresponds to the matched text term. The mask is annotated with the CVAT annotation platform [¶]. Additionally, the annotators provide a target text prompt with the instruction that i) it should be inconsistent with the original text but match the context that may mislead the readers; ii) it should not share semantic overlap with the original term, for instance, replacing "Chiwetel Ejiofor" with "Brad Pitt" rather than "a male actor". By utilizing the selected object region and annotated prompt as inputs to the DALL-E2 model, we obtain three manipulated images for each image-text pair. The second phase of the annotation process focuses on data cleaning to ensure the accuracy and coherence of the dataset. Each human annotator carefully follows three steps for quality checking purposes: (1) assessing the image quality of the generated images, (2) evaluating image-text inconsistencies, and (3) refining the region masks to establish ground truth for pixel-level inconsistency masks as the generated objects may have different shapes. To maintain the highest annotation quality, a rigorous cross-validation procedure was implemented within the team. This involved multiple assessments of each image-text pair by different annotators.

**Dataset Statistics.** The images in TIIL dataset have resolutions ranging from 256x396 to 3744x3744 pixels, and on average, the consistent mask covers 44.73% of the entire image area. There are a total of 7,101 consistent pairs in the dataset. Among them, 2,101 are composed of original images and text sourced from the Visual News dataset (Liu et al., 2020) (referred to as $\{I, T\}$ in Section 4 of the main paper). The remaining pairs consist of images generated by DALL-E2 (Ramesh et al., 2022) along with their corresponding text (referred to as $\{I_e, T_m\}$). The dataset also includes 7,138 inconsistent pairs, out of which 2,101 pairs consist of original images with manipulated text (referred to as $\{I, T_m\}$), while the remaining pairs consist of images generated by DALL-E2 (Ramesh et al., 2022) paired with the original text (referred to as $\{I_e, T\}$). Regarding manipulation regions, the dataset contains 3,015 annotated inconsistent regions of large size (greater than $200 \times 200$), 1,548 annotated inconsistent regions of medium size (ranging between $100 \times 100$ and $200 \times 200$), and 474 annotated inconsistent regions of small size (smaller than $100 \times 100$).

## B  ADDITIONAL ABLATION STUDIES

To demonstrate the impact of mask binarization threshold and text embedding alignment initialization methods on the performance, we offer additional ablation studies in this section.

**Text Embedding Alignment Initialization.** Instead of initializing the text embedding with random noise, we choose to initialize it with the embedding $E_0$ of the input text. In Table 6 of the main paper, we have provided a comparison of different embedding initialization methods in terms of mIoU. Here, we further present a comparison between random initialization and our $E_0$-based initialization using CLIP scores to explain our motivation. Table 7 demonstrates that the semantic similarity between inconsistent text embeddings (i.e., the input $E_0$ and the text embedding corresponding to the input

---

[§]https://github.com/explosion/spaCy
[¶]https://www.cvat.ai/

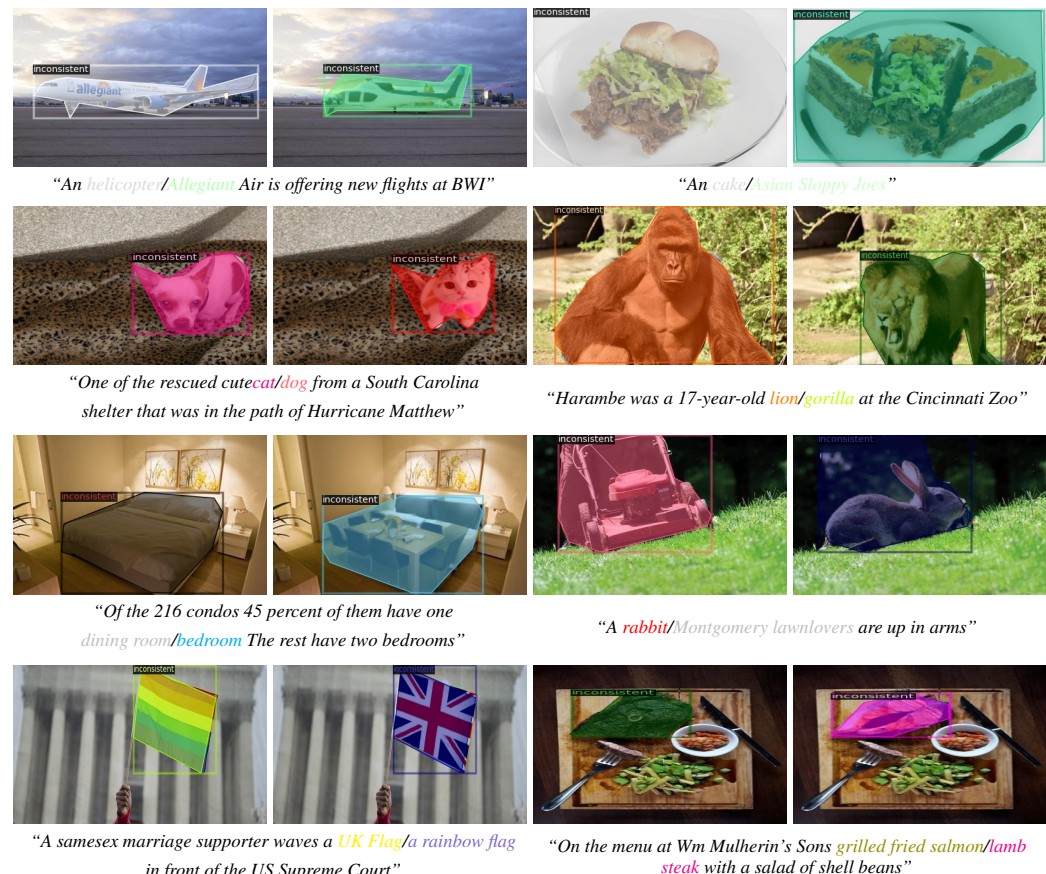

Figure 11: *Examples in TIIL dataset. Colored texts correspond to inconsistent regions of the same color in the image. The figure is best viewed in color.*

inconsistent image) is significantly higher than the similarity of random noise embeddings (i.e., the input $E_0$ and the randomly initialized embedding). This highlights the advantage of initializing with $E_0$, as it not only serves as a reference for optimizing $E$, but can also expedite the convergence during the alignment process.

|  | Random Init | Ours |
|---|---|---|
| CLIP Score | 0.19 | 80.98 |

Table 7: *Comparison of different text embedding initialization methods.*

**Comparison Between Image-manipulated and Text-manipulated Subsets.** As TIIL dataset is able to be categorized by two different kinds of manipulations: image-manipulated set and text-manipulated set, we further provide the performance of different methods on those two subsets for both localization (in Table 8) and detection (in Table 9). The text-changed samples achieved slightly worse performance than image-manipulated data, since the text changes (such as replacing a word or phrase) are less impactful than image changes (where various region sizes are manipulated), therefore more difficult to be detected.

|  | Image-changed | Text-changed |
|---|---|---|
| DetCLIP | 11.93 | 15.21 |
| GAE | 28.57 | 26.85 |
| Ours | 50.28 | 45.57 |

Table 8: *Comparison of the localization performance in different samples.*

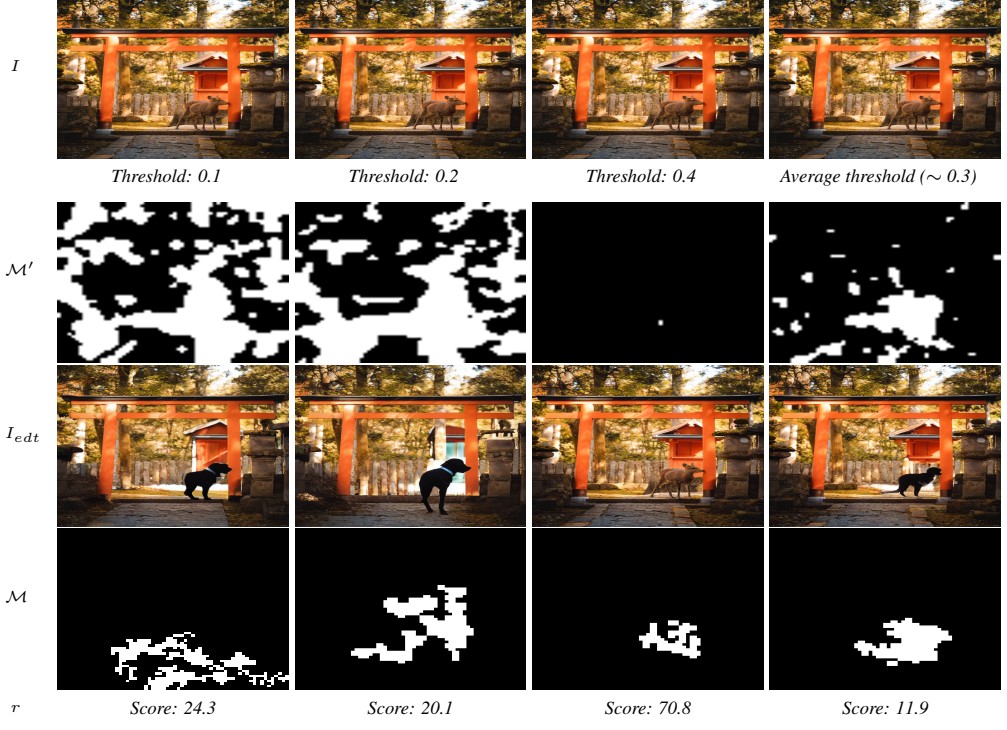

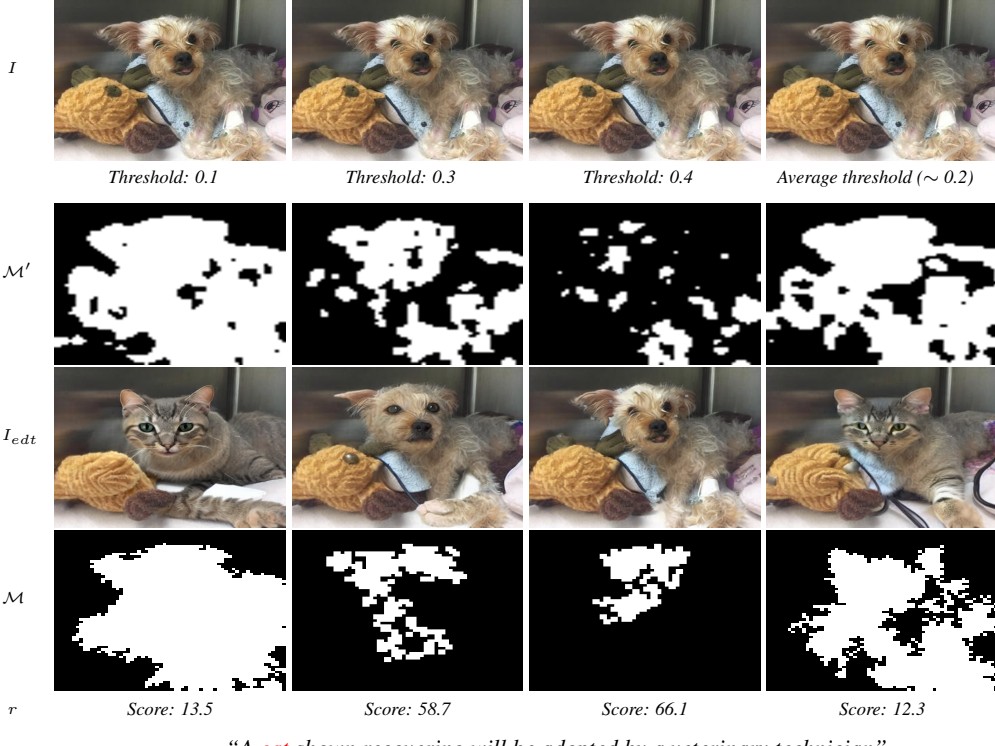

Figure 12: *Comparison of D-TIIL on TIIL examples with different thresholds.*

**Mask Threshold.** We compare four fixed threshold strategies with our average-based method, which uses the average values among the mask as the threshold for $\mathcal{M}'$ and $\mathcal{M}$. The results in Table 10

|  | AUC (image-changed) | ACC (image-changed) | AUC (text-changed) | ACC (text-changed) |
|---|---|---|---|---|
| CLIP | 86.50 | 79.11 | 78.58 | 71.87 |
| Ours | 88.80 | 81.25 | 82.29 | 74.70 |

Table 9: *Comparison of the detection performance in different samples.*

|  | 0.1 | 0.2 | 0.3 | 0.4 | Average (Ours) |
|---|---|---|---|---|---|
| mIoU (%) | 39.66 | 42.81 | 43.23 | 42.66 | 47.50 |

Table 10: *Comparison of different mask thresholds on a subset of TIIL with 1,000 image-text pairs.*

and Fig. 12 show that using a relatively smaller threshold results in a lower mIoU score and a larger predicted area, wherein the generated image $I_{edt}$ includes more "implicit" backgrounds. On the contrary, when using a larger threshold, there is an increase in the mIoU, but it can also lead to a smaller predicted inconsistency area. This is particularly evident when setting the threshold to 0.4, as depicted in Fig. 12. Our method surpasses the performance of fixed threshold strategies by utilizing an adaptive threshold for mask binarization.

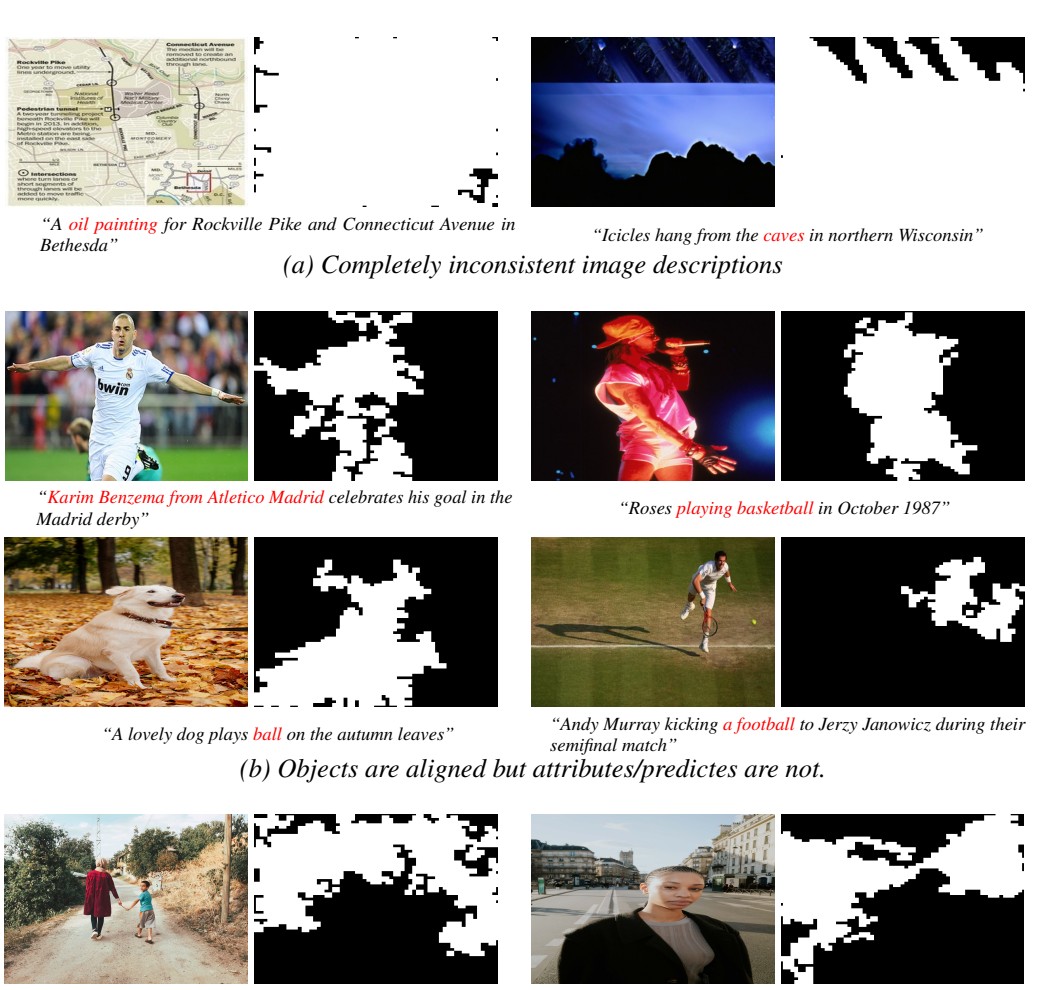

*"A oil painting for Rockville Pike and Connecticut Avenue in Bethesda"*

*"Icicles hang from the caves in northern Wisconsin"*

*(a) Completely inconsistent image descriptions*

*"Karim Benzema from Atletico Madrid celebrates his goal in the Madrid derby"*

*"Roses playing basketball in October 1987"*

*"A lovely dog plays ball on the autumn leaves"*

*"Andy Murray kicking a football to Jerzy Janowicz during their semifinal match"*

*(b) Objects are aligned but attributes/predictes are not.*

*"Woman and boy holding a stick while walking through the city"*

*"Portrait of woman on the forest"*

*(c) Backgrounds or scenarios are inconsistent.*

Figure 13: *Additional examples from D-TIIL on real-world news image-text pairs. The detected inconsistent text is highlighted as red.*

**Additional examples.** We include additional examples to cover different scenarios of inconsistencies in Fig. 13. Fig. 13 (a) shows the case that text and images are completely misaligned where the most area of the image is supposed to be masked; Fig. 13 (b) shows the case that the objects shared in the image and textual semantic space are well aligned but their attributes (e.g., actions, adjective) are inconsistent, the masks are supposed to cover the whole objects. Fig. 13 (c) contains more complex semantics inconsistent cases where the semantics inconsistency occurs in the background or the scene.

**Analysis of the learned representation.** D-TIIL has two alignment steps to iteratively align the image/text embeddings and filter out relevant semantic information with diffusion models. The well-aligned representations make it easier to identify and localize the semantic inconsistencies. Specifically, the learned representations are two parts, including $E_{aln}$ with aligned semantic space with the input image $I$, and $E_{dnt}$ with aligned semantic space with the input text embedding $E_0$. To show the effectiveness of our two alignment steps in learning the representation, we have provided the comparison of averaged cosine similarity scores on a subset of our dataset. We observed that the similarity between $E_{aln}$ and $I$ has increased by 9.2% compared to the similarity between $E_0$ and $I$ after the first step alignment. The similarity between $E_{dnt}$ and $I_{edt}$ is increased by 2.4% compared to the similarity between $E_0$ and $I_{edt}$ by the second step alignment. Note that 2.4% is not subtle since it only comes from the exclusion of distracting semantics from text embeddings.

