# OpenReview forum: "Exposing Text-Image Inconsistency Using Diffusion Models"
_ICLR.cc/2024/Conference — ICLR 2024 poster_

### Official Review · Reviewer_D7Cn · 2023-10-28

**Soundness:** 3 good
**Presentation:** 2 fair
**Contribution:** 3 good
**Rating:** 6
**Confidence:** 3

**Summary:**

This paper develops a new method, D-TIIL, to expose text-image inconsistency with the location of inconsistent image regions and words, which is quite commonly happening in T2I generation diffusion models. To achieve this, they introduce a new dataset, TIIL, for evaluating text-image inconsistency localization with pixel-level and word-level inconsistency annotations.

**Strengths:**

1. The dataset's contribution is commendable. Existing datasets lack the capacity to furnish evidence regarding inconsistencies occurring at both the image region and word levels, which is essential for evaluating D-TIIL (Diffusion-Based Text-to-Image Inconsistency Localization).

2. The problem addressed in this research is of significant importance. Previous methods have primarily focused on determining the presence of inconsistencies, whereas this paper introduces a novel approach to pinpointing the specific locations where these inconsistencies occur.

**Weaknesses:**

1. It would be valuable to explore whether this method could be extended to evaluate other text-to-image (T2I) augmentation techniques (i.e., [1-3]). Given the abundance of research on generating images based on textual prompts, applying this method for evaluation purposes could have a broader impact and contribute significantly to the field.

2. Are there alternative evaluation metrics to assess the correspondence between text and images? Based on my experience with CLIP scores, it may not consistently capture performance accurately in various scenarios.

[1] Attend-and-Excite: Attention-Based Semantic Guidance for Text-to-Image Diffusion Models. SIGGRAPH 2023

[2] Improving Sample Quality of Diffusion Models Using Self-Attention Guidance. ICCV 2023

[3] Expressive Text-to-Image Generation with Rich Text. ICCV 2023

**Questions:**

As mentioned in the above weakness, I would appreciate seeing the proposed method applied more extensively in evaluation. The inclusion of evaluation metrics beyond CLIP scores could enhance the robustness and confidence of this paper.

---

> ### Author Response · Authors · 2023-11-15
> **Rebuttal by Authors**
>
> **Q1. Could this method be extended to evaluate other text-to-image (T2I) augmentation techniques?**
>
> Thanks for the suggestion. Running these experiments takes time so we cannot do it within the period of ICLR review. Technically, our method is not restricted to a specific text-to-image diffusion model, and can be extended to other most recent diffusion models and their variants.
>
> **Q2. Are there alternative evaluation metrics to assess the correspondence between text and images other than CLIP scores?**
>
> Following existing text-image correspondence evaluation methods [1-4], we used the CLIP score to assess the correspondence between text and images. It is worth noting that the CLIP score is only used to analyze the inconsistency levels of our dataset in Table 1 and evaluate the text embedding initialization effectiveness in Table 7, it might work well on our dataset as most of the inconsistencies in our dataset are in the object-level but not scenarios-level. We did not use the CLIP score in the text-image inconsistency localization pipeline, which will not influence the localization performance.
>
> [1] Meng, Chenlin, et al. "On distillation of guided diffusion models." Proceedings of the IEEE/CVF Conference on Computer Vision and Pattern Recognition. 2023.
>
> [2] Saharia, Chitwan, et al. "Photorealistic text-to-image diffusion models with deep language understanding." Advances in Neural Information Processing Systems 35 (2022): 36479-36494.
>
> [3] Couairon, Guillaume, et al. "Diffedit: Diffusion-based semantic image editing with mask guidance." ICLR. 2023.
>
> [4] Blattmann, Andreas, et al. "Retrieval-augmented diffusion models." Advances in Neural Information Processing Systems 35 (2022): 15309-15324.

---

### Official Review · Reviewer_4tYr · 2023-10-31

**Soundness:** 2 fair
**Presentation:** 3 good
**Contribution:** 2 fair
**Rating:** 5
**Confidence:** 4

**Summary:**

The authors propose D-TIIL (Diffusion-based Text-Image Inconsistency Localization), a system for automatically identifying and explaining text-image inconsistencies. D-TILL uses text-image diffusion models to locate semantic inconsistencies in text-image pairs. Diffusion models trained on large datasets filter out irrelevant information and incorporate background knowledge to identify inconsistencies. In addition, D-TIIL uses text embeddings and modified image regions to visualize these inconsistencies.
To evaluate the effectiveness of D-TIIL, the authors also introduce a new dataset (TIIL) with 14K consistent and inconsistent text-image pairs.

**Strengths:**

•	The paper is well written and well structured
•	The problem and the related work are well introduced
•	The framework is explained in detail
•	The idea to build consistency scores between stable diffusion and the original image is interesting.

**Weaknesses:**

•	The general theoretical idea behind the approach lacks clearity
•	The real-world application is not very clear, e.g. wrong labels have a different type of mislabeling than just objects that are swapped
•	Sensitivity to threshold highly influences M and the consistency score

With D-TIIL, the authors have presented an interesting method for using diffusion models to evaluate the consistency of image-text pairs.
However, the utility of the method is not fully evaluated in detail. Deeper insights into why this approach works are lacking. In addition, it would be nice to see how the approach works on other datasets where the labeling is just mixed up or misleading.
In addition, I would recommend for ICLR to investigate the method in more detail in terms of learned representations.

The paper is well written and has some interesting ideas, e.g. the usage of diffusion models for detecting image-text inconsistency. The method and the dataset, both are valuable. However, to be accepted in ICLR I would expect more and deeper investigations about the method and the dataset. What is learned, what are short comings?
There are some doubts, such that the model could be sensitive to the DALE generated part instead being sensitive to the text-image inconsistency. Experiments are missing that evaluate the underlying behavior. Moreover, a second evaluation on another dataset with more established baselines would be preferable to proof some of the assumptions, advantages and shortcomings of the method.

**Questions:**

•	How does the approach perform on completely wrong image descriptions?
o	Is the whole image masked?
•	Is the model sensitive to the image part generated by DALE and not to the parts which do not correspondent to the text?
o	Is there an experiment that can proof that?
o	Maybe regenerate the image for the dataset also with the right semantic class?
•	Is there another dataset where the method could be compared also to other baselines?

---

> ### Author Response · Authors · 2023-11-15
> **Rebuttal by Authors**
>
> **Q1. The general theoretical idea behind the approach lacks clearity.**
>
> Thank you for your suggestion on the paper. We will improve the writing in the methodology part to make it more readable. The theoretical idea behind the approach is to employ the text-to-image diffusion models as “omniscient” agents to align the image-text latent space so that our proposed two alignment steps are able to filter out irrelevant information and incorporate background knowledge to identify inconsistencies.
>
> **Q2. The real-world application is not very clear.**
>
> We describe the problem that this paper solves in the Introduction. Our method aims to expose misinformation on social media and the Internet created by juxtaposing images with texts that do not accurately reﬂect the image’s original meaning or intention.
>
> **Q3. Sensitivity to mask threshold**.
>
> We agree that the mask threshold influences the performance. Therefore, our method designs a sample-specific threshold based on the average values among the mask instead of using a fixed threshold for all samples. In Table 10 and Figure 12 of the Appendix, we conducted a comparison with four fixed threshold strategies to show the effectiveness of our mask threshold setting method.
>
> **Q4. Deeper insights into why this approach works are lacking.**
>
> Understanding the semantic connection between text and image is the key to solving the text-image inconsistency problem, and our method is the first to explore generative AI models for this purpose. Our results have revealed two aspects that are not known previously in the literature. First, we use the large-scale text-to-image diffusion models as a foundation model with extensive background knowledge to effectively align the text and image semantics. Second, we design two alignment steps to iteratively align the image/text (embeddings) and filter out relevant semantic information with diffusion models. Thus, we obtain knowledge-shared text embeddings from both the input image and text, making it easier to identify semantic inconsistencies.
>
> **Q5. Lack investigate the method in more detail in terms of learned representations.**
>
> We have included the investigation of the learned representations in the revised manuscript of paragraph “Analysis of the learned representation” in the Appendix.
>
> **Q6. How the approach works on other datasets,is there another dataset where the method could be compared?**
>
> To the best of our knowledge, the proposed TIIL dataset is the first text-image inconsistency localization dataset with pixel-level and word-level inconsistency annotations. The proposed TIIL dataset is based on content from diverse real-world news stories and a wide range of modifications in inconsistent regions, including local inconsistencies and global swapping based on mixed-up pairs.
>
> **Q7. Could the model be sensitive to the DALLE generated part instead being sensitive to the text-image inconsistency?**
>
> Our method is not sensitive to the part that DALL-E generated. As shown in the Appendix, we compared the performance of our method with baselines on Image-manipulated and Text-manipulated subsets in Tables 8 and 9. The results show the effectiveness and superiority of our method in both DALL-E generated inconsistent samples and non-DALL-E manipulated samples.
>
> **Q8 What are the shortcomings of proposed methods?**
>
> As discussed in Section 5.4 on failure cases, one shortcoming of our method is that given the limited prior knowledge of the diffusion model we used, our model may not effectively handle the inconsistencies with respect to specific external knowledge.
>
> **Q9. How does the approach perform on completely wrong image descriptions?**
>
> We have added examples with completely wrong image descriptions from our TIIL dataset in the Appendix Fig.13 (a) of the revised manuscript. Our method can provide a whole image mask for this kind of inconsistency samples.

---

### Official Review · Reviewer_LEKC · 2023-10-31

**Soundness:** 3 good
**Presentation:** 2 fair
**Contribution:** 2 fair
**Rating:** 5
**Confidence:** 3

**Summary:**

This paper presents D-TIIL for identifying and localizing inconsistencies between text and images.
A new dataset, TIIL, containing 14K consistent and inconsistent text-image pairs, is introduced for evaluating the method. The D-TIIL outperforms existing approaches in terms of Accuracy scores and demonstrates more explainable results. In a nutshell, the paper offers a scalable and evidence-based approach to identify and localize the text-image inconsistency. However, it also acknowledges the potential misuse of the method for creating deceptive text-image pairs and suggests improving the algorithm and restricting access.

**Strengths:**

1. Originality: The paper introduces a novel method, D-TIIL, that exposes text-image inconsistency with the location of inconsistent image regions and words. Also, the new TIIL dataset is the first dataset with pixel-level and word-level inconsistency features that provide fine-grained and reliable inconsistency.

2. Quality: The D-TIIL and TIIL dataset generation are thoroughly described. The paper also provides a comprehensive comparison of the proposed method with existing approaches.

3. Clarity: The paper is well-structured and clearly written. The method is explained in detail and the experiment results are presented in an understandable manner.

4. Significance: The D-TIIL method improves the accuracy of inconsistency detection and provides more explainable results. The introduction of the diffusion model makes it possible to align text and images in a latent and joint representation space to discount irrelevant information and incorporate broader knowledge.

**Weaknesses:**

1. The paper acknowledges that the D-TIIL may struggle with inconsistencies with respect to specific external knowledge, and this could reduce the effectiveness of the method in real-world application.

2. The D-TIIL method relies heavily on the text-to-image diffusion models and benefits a lot from the semantic space that is already well aligned. This dependence could limit the generalizability of the proposed method.

3. There are some confusing details in the method description section.

4. In the comparison of methods, the reasons why D-TIIL is superior are not discussed and analyzed in detail, and the potential solutions for the failure cases are not provided.

5. More specific discussions and measures could be included to prevent potential abuse rather than simply restricting access.

**Questions:**

1. Regarding Step 3 in Section 3 METHOD, the proposed E_{dnt} and descriptions like “include extra implicit information from the images and excludes additional implicit information that only appears in the text” raise doubts about the effectiveness of the process of the “text denoising”. Such “text denoising” seems to be too idealistic. In Section 5.4, for example, there is the failure case of the word "office". This leads to the bold suspicion that the D-TIIL method is only valid for simple objects, but not for backgrounds or objects that contain more complex semantics.

2. Also, the high dependency on the diffusion model affects the generalizability of the method. If text and image are not well aligned on the latent space, the validity of the method will be more affected. Semantic entanglement can also exist.

3. Regarding Step 4 in Section 3 METHOD, the descriptions like “We then compute the cosine similarity score between this image embedding and the input text embedding” are confusing to the readers.

4. In the Data Generation part of Section 4 TILL DATASET, T_{m} is unknown where it comes from, is it manually designed?

---

> ### Author Response · Authors · 2023-11-15
> **Rebuttal by Authors**
>
> **Q1. D-TIIL may struggle with inconsistencies with respect to specific external knowledge which could reduce the effectiveness of the method in real-world application.**
>
> With the rapid development of text-to-image (T2I) diffusion models, our D-TIIL would be more generalizable to handle different types of real-world cases, such as using domain-specific diffusion models as mentioned in the Conclusion or more powerful diffusion models like the Stable Diffusion XL.
>
> **Q2. The dependence with diffusion model could limit the generalizability of the proposed method, especially when text and image are not well aligned on the latent space.**
>
> Our method leverages the semantic alignment performance of T2I diffusion models, but would fail for complex cases that current diffusion models cannot handle. However, our method is not restricted to a specific T2I diffusion model and can be implemented and adapted to the most recent diffusion models to enhance its generalizability further. In terms of the generalizability to the capacity for identifying various degrees of inconsistency in text-image pairs, our current method can address not only subtle inconsistencies with local manipulations but also whole swapping and mix-ups. We provided some examples of such scenarios in Fig. 13 (a) and (c) of the revised manuscript.
>
> **Q3. There are some confusing details in the method description section.**
>
> Thanks for pointing this out. We will improve the writing of the methodology part in the finalized version of the paper.
>
> **Q4. The reasons why D-TIIL is superior are not discussed and analyzed in detail, and the potential solutions for the failure cases are not provided.**
>
> Our D-TIIL outperforms baselines in comparison experiments due to the following two reasons. Unlike the baseline DetCLIP using object segmentation to compare CLIP similarity or GAE using attention heatmap for CLIP, our method leverages the T2I diffusion models to learn the semantic connections between textual and visual information. More importantly, instead of directly comparing the text and image embeddings, D-TIIL relies on two-step alignment to iteratively exclude irrelevant information and obtain knowledge-shared representations from two modalities and then directly exposes the inconsistencies.
> The potential solutions for failure cases involve using more powerful diffusion models such as the most recent T2I diffusion models or domain-specific diffusion models, as discussed in the Conclusion section of the submission.
>
>
> **Q5. More specific discussions and measures could be included to prevent potential abuse rather than simply restricting access.**
>
> We will release our code as open-source with the condition that it “must not distribute harmful, offensive, dehumanizing content or otherwise harmful representations of people or their environments, cultures, religions, etc. produced with the model weights”.
>
> **Q6. The term “text denoising” is too idealistic and D-TIIL may not localize the inconsistency that backgrounds or objects that contain more complex semantics.**
>
> Thanks for the suggestion, we will consider using the term “text alignment” other than “text denoising”. This process actually reduces the distance between the image and text semantic space as shown in the "Analysis of the learned representation" section of the Appendix in the revised manuscript.
> Our method can handle inconsistencies in not only simple objects, but also the backgrounds, and attributes of objects. We have included such examples in Fig. 13 (c) of the Appendix.
>
> **Q7. The descriptions of Step 4 in Method is not clear.**
>
> Thanks for pointing this out. We have modified the statement as follows: “We then compute the cosine similarity score between the CLIP image embedding of the masked image and the input text embedding E_0 as the consistency score.”.
>
> **Q8. IN Section 4 DATASET, T_{m} is unknown.**
>
> T_{m} is the alerted text based on human annotations. We have clarified this in Section 4. Thanks for pointing this out.

---

> > ### Comment · Reviewer_LEKC · 2023-11-22
> > **Post-rebuttal comments.**
> >
> > Dear Authors,
> >
> > Thank you for your detailed response to my comment.
> > Your explanation enhances my understanding of your work.
> > However, I would like to emphasize that my primary concern remains regarding the limitations of the approach.
> > Specifically, the effectiveness of the approach seems heavily dependent on the well-trained T2I diffusion models, where the textual and semantic spaces are already well aligned. For diffusion models that are not as well-trained, the approach will not work. This dependency may limit its applicability and generalizability, thus diminishing the contribution of your work.
> >
> > --Reviewer LEKC

---

> > > ### Author Response · Authors · 2023-11-22
> > > **Official Comment by Authors**
> > >
> > > Thank you for your comments. We agree with the reviewer that using diffusion models which are not well-trained would limit the applicability of our method. Therefore, we build our method upon the well-trained diffusion models, for example, Stable Diffusion, that was trained on pairs of images and captions taken from LAION-5B with 5 billion image-text pairs. The semantic space alignment of Stable Diffusion has been shown effective in a variety of tasks, such as text-guided image editing [1,2], image-text matching [3], and image captioning [4]. As such, the generalizability of the proposed method would benefit from the broad knowledge and semantic alignment of diffusion models. The training of diffusion models and enhancement of semantic alignment ability of diffusion models is an interesting and different topic which many researchers are working on. Our method takes well-trained diffusion models as foundation models, and could always be updated to a diffusion model that has better alignment for better generalizability.
> > >
> > > ---
> > >
> > > Reference:
> > >
> > > [1] Couairon, Guillaume, et al. “Diffedit: Diffusion-based semantic image editing with mask guidance.” ICLR.
> > > 2023.
> > >
> > > [2] Mokady, Ron, et al. “Null-text inversion for editing real images using guided diffusion models.” CVPR, 2023.
> > >
> > > [3] Krojer, Benno, et al. “Are diffusion models vision-and-language reasoners?.” NeurIPS, 2023.
> > >
> > > [4] Xiao, Changrong, Sean Xin Xu, and Kunpeng Zhang. “Multimodal Data Augmentation for Image
> > > Captioning using Diffusion Models.” arXiv preprint arXiv:2305.01855 (2023).

---

### Official Review · Reviewer_qTYR · 2023-10-31

**Soundness:** 3 good
**Presentation:** 2 fair
**Contribution:** 4 excellent
**Rating:** 8
**Confidence:** 3

**Summary:**

This paper studies how to detect image-text inconsistency with diffusion models. More specifically, the author designed a pipeline that iteratively use diffusion models to edit the text and images in the image-text pairs to gradually optimize a mask that can point out where the inconsistency come from. This task is interesting and meaningful for misinformation detection, as it provides interpretable prediction results. To evaluate the proposed method, the authors collected a dataset containing image-text pairs and their inconsistency masks. Experiments shows that the proposed method outperforms baselines and gives explanable prediction on the inconsistency.

**Strengths:**

1. The task studied in this paper is meaningful.

2. The dataset that they collected is contributive to the community.

3. The method is novel.

**Weaknesses:**

1. The writing is not very good. I read the methodology part several hours to understand their pipeline.

2. The idea is well justified for the inconsistency of object alignment. But what if the predicate is not aligned, i.e. the person is correct but the action is not?

**Questions:**

How does the annotation and the model handles predicates?

---

> ### Author Response · Authors · 2023-11-15
> **Rebuttal by Authors**
>
> **Q1. The writing needs to be improved.**
>
> Thank you for your suggestion on the paper. We will improve the writing in the finalized version of this manuscript and pay special attention to the methodology part to make it more readable.
>
> **Q2. How does the annotation and the model handle predicates inconsistencies?**
>
> This is a good point. Our method can well handle inconsistencies in not only objects but also predicates and adjectives. We have added examples in the Appendix Fig. 13 of the revised manuscript. In terms of dataset construction, the annotators are instructed to select and edit object-term pairs from image and text, which include editing in objects, scenes, and attributes based on predicates or adjectives. For example, a singing man is edited with the prompt “a man playing basketball”, and a yellow cat can be changed with the prompt “a red cat”.

---

### Author Response · Authors · 2023-11-15
**General Comments**

We thank all the reviewers for their time, insightful suggestions, and valuable comments. We are grateful for the positive recognition of the reviewers that our idea and task are interesting and meaningful (Reviewers qTYR, 4tYr, and D7Cn), the method is novel and provides interpretable evidence (Reviewers qTYR and LEKC), the paper is well written (Reviewers LEKC and 4tYr), and our dataset is contributive to the community (Reviewers qTYR and D7Cn).

We have responded to each reviewer's comments in detail below. A revised version of the manuscript has been uploaded. We hope our response and rebuttal revision will address the reviewers' concerns.

---

### Meta-Review · Area_Chair_DiDR · 2023-12-05

**Metareview:**

The reviewers appreciate the method, dataset and writing (with some exceptions). While there are some questions, none seem critical, and the authors attempted to address them.

**Justification For Why Not Higher Score:**

Only one 8, and from a very brief review

**Justification For Why Not Lower Score:**

Many scores are borderline, but list numerous meaningful strengths

---

### Decision · Program_Chairs · 2024-01-16

Accept (poster)